Genome-wide analysis suggests high level of microsynteny and purifying selection affect the evolution of EIN3/EIL family in Rosaceae

Cao Yunpeng 1
Han Yahui 2
Meng Dandan 1
Li Dahui 1
Jin Qing 1
Lin Yi 1
Cai Yongping ypcaiah@163.com 1
1 College of Life Sciences, Anhui Agricultural University , Hefei , China
2 State Key Laboratory of Tea Plant Biology and Utilization, Anhui Agricultural University , Hefei , China
Röder Marion
Electronic publication date: 2017 May 31
Publication date: 2017
Volume: 5
Electronic Location ID: e3400
Received 2017 Jan 20; Accepted 2017 May 10
Copyright: ©2017 Cao et al.
Copyright year: 2017
Copyright holder: Cao et al.
License: This is an open access article distributed under the terms of the Creative Commons Attribution License, which permits unrestricted use, distribution, reproduction and adaptation in any medium and for any purpose provided that it is properly attributed. For attribution, the original author(s), title, publication source (PeerJ) and either DOI or URL of the article must be cited.
License URL: https://creativecommons.org/licenses/by/4.0/

Keywords: Rosaceae, EIN3/EIL, Microsynteny, Purifying selection, qRT-PCR

Funding: National Natural Science Foundation of China 31640068 2017 Graduate innovation fund of Anhui Agriculture University #2017yjs-31 This study was supported by the National Natural Science Foundation of China (grant 31640068) and 2017 Graduate innovation fund of Anhui Agriculture University (grant #2017yjs-31). The funders had no role in study design, data collection and analysis, decision to publish, or preparation of the manuscript.

==============================
The ethylene-insensitive3/ethylene-insensitive3-like (EIN3/EIL) proteins are a type of nuclear-localized protein with DNA-binding activity in plants. Although the EIN3/EIL gene family has been studied in several plant species, little is known about comprehensive study of the EIN3/EIL gene family in Rosaceae. In this study, ten, five, four, and five EIN3/EIL genes were identified in the genomes of pear (Pyrus bretschneideri), mei (Prunus mume), peach (Prunus persica) and strawberry (Fragaria vesca), respectively. Twenty-eight chromosomal segments of EIL/EIN3 gene family were found in four Rosaceae species, and these segments could form seven orthologous or paralogous groups based on interspecies or intraspecies gene colinearity (microsynteny) analysis. Moreover, the highly conserved regions of microsynteny were found in four Rosaceae species. Subsequently it was found that both whole genome duplication and tandem duplication events significantly contributed to the EIL/EIN3 gene family expansion. Gene expression analysis of the EIL/EIN3 genes in the pear revealed subfunctionalization for several PbEIL genes derived from whole genome duplication. It is noteworthy that according to environmental selection pressure analysis, the strong purifying selection should dominate the maintenance of the EIL/EIN3 gene family in four Rosaceae species. These results provided useful information on Rosaceae EIL/EIN3 genes, as well as insights into the evolution of this gene family in four Rosaceae species. Furthermore, high level of microsynteny in the four Rosaceae plants suggested that a large-scale genome duplication event in the EIL/EIN3 gene family was predated to speciation.

Introduction

Rosaceae species such as pear (Pyrus bretschneideri), mei (Prunus mume), peach (Prunus persica) and strawberry (Fragaria vesca) are important perennial trees cultivated for the commercial production of fruits available worldwide. According to previous studies, the genomes of strawberry (X = 7), mei (X = 8), peach (X = 8), and pear (X = 17) shared an ancestor, which had nine pairs of chromosomes (Shulaev et al., 2011; Verde et al., 2013; Wu et al., 2012; Zhang et al., 2012). Recently, the researchers confirmed that chromosome inversions, fusions, and translocations played an important role in the evolution of the Rosaceae genome (Illa et al., 2011). Some extant “diploid” species of Rosaceae family are originated from their polyploid ancestors, others are actually thought to be true polyploids (Wendel, 2000). These studies indicate that the diploid species in Rosaceae have evolved with a complex history. There are several gene families which share highly conserved genome sequences with each other among the related species of family Rosaceae, as well as other taxonomic families. In this study, the EIL/EIN3 gene family was selected to investigate the specific evolutionary relationships among the related species of family Rosaceae.

The EIN3/EIL gene family is a relatively small one in higher plants. Some EIN3/EIL genes have been isolated from Arabidopsis thaliana (Chao et al., 1997), tobacco (Kosugi & Ohashi, 2000; Rieu, Mariani & Weterings, 2003), banana (Jourda et al., 2014), tomato (Tieman et al., 2001; Yokotani et al., 2003) and rice (Hiraga et al., 2009; Mao et al., 2006). These plant-specific EIN3/EIL proteins are located in the nuclei, with the highly conserved amino acid sequences at the N-termini, including several important structural features, such as acidic amino acid regions, proline-rich regions and 5-basic amino acid clusters (BD I–V) (Chao et al., 1997). Compared to the N-terminal sequences, the conservation of their C-termini is lower. For example, it was found that although asparagine-enriched regions or glutamine-enriched regions were commonly distributed within the C-terminal sequences of EIN3/EILs in Arabidopsis, carnation and mung beans (Chao et al., 1997; Lee & Kim, 2003; Waki et al., 2001), they did not widely exist in other EIN3/EIL members, such as tobacco NtEILs (Rieu, Mariani & Weterings, 2003).

Functions of the EIN3/EIL gene family have been studied in several plants, such as Hevea brasiliensis (Yang et al., 2015) and tomato (Tieman et al., 2001; Yokotani et al., 2003). Recently, research on the application of comparative genome in the analysis of evolution and function of gene family have been reported. For example, based on the comparative genomic analysis, Wang et al. (2015) explored the evolution and functional differences of WRKY type-III transcription factor family of poplar, grape, Arabidopsis and rice. Jing et al. (2016) explored the evolution of WRKY I subfamily in Gramineae. However, there is still lack of specific evolutionary relationships of the EIN3/EIL gene family in Rosaceae. To address this question, the evolutionary relationships and gene duplication events of EIN3/EIL genes from Rosaceae species, including pear (Pyrus bretschneideri), mei (Prunus mume), peach (Prunus persica) and strawberry (Fragaria vesca), were analyzed, based on their phylogenetic relationships, microsynteny and environmental selection pressures analysis. In addition, the expression patterns of pear EIN3/EIL genes were investigated on a variety of organs/tissues including fruits at several developmental stages. The results obtained from this study provided valuable information about EIN3/EIL genes that will aid future functional research involved in many important biological processes of this important gene family in flowering plants, especially in the pear.

Materials and Methods

Sequence identification and collection

The genome data of four Rosaceae species were obtained from their respective genome sequence websites: Pyrus bretschneideri from the GigaDB database (http://gigadb.org/site/index); Prunus mume from the Genome Database for Rosaceae (http://www.rosaceae.org/); Prunus persica from the Phytozome database (https://phytozome.jgi.doe.gov/pz/portal.html) and Fragaria vesca from the Joint Genome Institute (http://www.jgi.doe.gov/). The Hidden Markov Model (HMM) profiles of EIN3/EIL domain (PF04873) (Chang & Shockey, 1999; Chao et al., 1997) were obtained from the Pfam database (http://pfam.xfam.org) (Finn et al., 2006). The EIN3/EIL domain was used as query sequences to identify EIN3/EIL genes in four Rosaceae species by using DNAtools software (E-value < 0.001). To verify the EIN3/EIL genes in four Rosaceae genomes, all putative proteins were validated by searching for the EIN3/EIL domain using the InterPro online tool (http://www.ebi.ac.uk/interpro/) (Zdobnov & Apweiler, 2001) and SMART database (http://smart.embl-heidelberg.de/) (Letunic, Doerks & Bork, 2012). In our study, only the EIN3/EIL domain-containing sequences were retained.

Chromosomal location of EIN3/EIL genes

The genome annotation information was collected from GigaDB database (http://gigadb.org/site/index), Genome Database for Rosaceae (http://www.rosaceae.org/), Phytozome database (https://phytozome.jgi.doe.gov/pz/portal.html) and Joint Genome Institude (http://www.jgi.doe.gov/), respectively. Subsequently, the MapInspect software (http://mapinspect.software.informer.com/) was used for data visualization.

Gene structure and motif analysis

The exon-intron structure of each EIN3/EIL gene was determined by alignment of its CDS and genomic DNA sequence. Then a diagram was constructed using the Structure Display Server website (Hu et al., 2014). Subsequently, the Online MEME server was used to screen the conserved motifs encoded by EIN3/EIL genes. Additionally, the Pfam website (Punta et al., 2012) and SMART tools (Letunic, Doerks & Bork, 2012) were used to annotate these structural motifs.

Phylogenetic analysis of EIN3/EIL genes

EIN3/EIL sequences were aligned using ClustalX version 1.83 (Thompson et al., 1997) and evolutionary relationships were inferred by analyzing an unrooted phylogenetic tree using MEGA 5 and neighbor-joining (NJ) method (Tamura et al., 2011) with the following parameters: poisson correction, pairwise deletion and 1,000 bootstrap replicate.

Microsynteny analysis

In order to reveal the sequence features of the EIN3/EIL gene-containing regions, microsynteny analysis was performed across the four Rosaceae species using MCScanx (Multiple Collinearity Scan toolkit) (Wang et al., 2012) with the gene identifier file, the gene list file and the coding sequence file. Subsequently, a syntenic block was defined as a region containing three or more conserved homologs which were located within 100-kb downstream and upstream of protein-coding sequences.

Environmental selection pressure analysis

The nucleotide coding sequences from segmentally duplicated pairs were aligned by Clustal X (Thompson et al., 1997). Then DnaSP (version 5.10) was used to calculate the nonsynonymous (Ka) and synonymous (Ks) substitution rates of the homologues (Librado & Rozas, 2009). For each pair of duplicated regions, we estimated the mean Ks values of the flanking conserved genes for individual homologs. To further understand the selective pressure experienced by EIN3/EIL genes, Ka, Ks and Ka/Ks ratios were estimated using sliding window (with parameters: window size, 150 bp; step size, 9 bp) over the entire aligned length (Cao et al., 2016a; Han et al., 2016).

EIN3/EIL gene expression analysis in pear different tissues

To verify the expression patterns of EIN3/EIL genes, qRT-PCR analysis was carried out. The first-strand cDNA was synthesized with Oligo18dT primer (Table S1) by using M-MLV reverse transcriptase (TakaRa, Japan) following the manufacture introduction. The TransStart Tip Green qPCR SuperMix (TransGen Biotech, China) with SYBR Green I as the fluorescent dye was used for the qPCR, employing a Bio-rad CFX96 Touch™Deep Well Real-Time PCR Detection system (BioRad, USA). The transcript level relative to the Pyrus tubulin gene (Wu et al., 2012) was estimated according to a previous workflow (Cao et al., 2016a; Cao et al., 2016b). For each sample, three replicates were set up in parallel experiments.

Results and Discussion

Identification of EIN3/EIL genes in Rosaceae

The genome data of pear (P. bretschneideri), mei (P. mume), peach (P. persica) and strawberry (F. vesca) were recently published, respectively (Shulaev et al., 2011; Verde et al., 2013; Wu et al., 2012; Zhang et al., 2012). To identify the members of the EIN3/EIL gene families in these species, EIN3/EIL specific domain (PF04873) was used to perform Blastp searches of the local protein databases. Sequences identified were verified for EIN3/EIL domains through SMART database and InterPro online tool. In total 24 of EIN3/EIL genes were identified, including ten in pear, four in peach, five in mei and five in strawberry, and named as PbEIL1-PbEIL10, PpEIL1-PpEIL4, PmEIL1-PmEIL5 and FvEIL1-FvEIL5, according to their locations in chromosome, respectively (Table 1 and Fig. 1). This result suggested that EIN3/EIL gene family was relatively small compared to other gene families in the studied species. Similar indication was also reported by the previous studies in which six, five, four, six and 17 EIN3/EIL genes were found in Arabidopsis thaliana (Chao et al., 1997; Guo & Ecker, 2004), tobacco (Kosugi & Ohashi, 2000; Rieu, Mariani & Weterings, 2003), tomato (Tieman et al., 2001; Yokotani et al., 2003), rice (Hiraga et al., 2009; Mao et al., 2006) and banana (Jourda et al., 2014), respectively. Furthermore, it was found that the genome sizes and number of EIN3/EIL gene family members appeared not to have a direct relevance. For example, although there was no significant variety in genome size of pear (271.9 Mb) and strawberry (240 Mb), the number of EIN3/EIL genes obviously changed. Contrarily, the number of EIN3/EIL genes of the peach (224.6 Mb) and strawberry (240 Mb) had a corresponding relationship with their genome size. Remarkably, compared with those in peach, mei and strawberry, the numbers of EIN3/EIL genes in pear were found to be almost doubled. Moreover, the chromosome numbers of peach, mei and strawberry are 16, 16 and 14, respectively (Shulaev et al., 2011; Verde et al., 2013; Wu et al., 2012; Zhang et al., 2012), whereas the chromosome number of pear is 34, indicating that the EIN3/EIL gene family has undergone an expansion corresponding to the variation in chromosome number. However, a recent whole genome duplication event (30–45 million years ago) (Wu et al., 2012) that occurred in pear but not in peach, mei and strawberry probably contributed to the expansion of EIN3/EIL gene family in the pear.

Table 1 List of EIN3/EIL genes identified in pear, peach, mei and strawberry.

Name	Gene model	Chromosme	5′ end	3′ end	
FvEIL1	mrna25474.1	Chr1	17653671	17655527	
FvEIL2	mrna16361.1	Chr1	18891616	18892965	
FvEIL3	mrna20650.1	Chr3	29248944	29253704	
FvEIL4	mrna00379.1	Chr7	290967	292781	
FvEIL5	mrna00392.1	Chr7	349495	351202	
PmEIL1	Pm001950	Chr1	15239829	15241073	
PmEIL2	Pm002057	Chr1	16248534	16250294	
PmEIL3	Pm017009	Chr5	6907428	6909233	
PmEIL4	Pm017011	Chr5	6933006	6934874	
PmEIL5	Pm028171	scaffold103	1235430	1246520	
PpEIL1	ppa003493m	Chr2	5516222	5518429	
PpEIL2	ppa003550m	Chr2	5549949	5552334	
PpEIL3	ppa003113m	Chr6	3882268	3885188	
PpEIL4	ppa016118m	Chr6	16979366	16982360	
PbEIL1	Pbr024739.1	Chr2	8493409	8495211	
PbEIL2	Pbr024740.1	Chr2	8506285	8508129	
PbEIL3	Pbr000646.1	Chr3	18718500	18721454	
PbEIL4	Pbr026603.1	Chr8	3598382	3602224	
PbEIL5	Pbr004535.1	Chr11	22794386	22798326	
PbEIL6	Pbr033210.1	Chr15	31009957	31014042	
PbEIL7	Pbr010447.1	scaffold170.2.1	239188	246113	
PbEIL8	Pbr010448.1	scaffold170.2.1	259361	262132	
PbEIL9	Pbr022557.1	scaffold341.0	56672	57973	
PbEIL10	Pbr039294.1	scaffold837.0	82840	84144	
Notes.

Pear gene models are found in the GigaDB Genome database; mei and peach gene models are found in the Rosaceae Genome Database; strawberry gene models are found in the Phytozome database.

Figure 1 Chromosomal location of EIN3/EIL genes in the genomes of strawberry (A), mei (B), pear (C) and peach (D).

The distribution of EIN3/EIL genes among the chromosomes in each species was diverse. The chromosome number was represented at the top of each chromosome. The left scale indicates the megabases (Mb).

To determine the distribution of EIN3/EIL genes on chromosomes among pear, peach, mei and strawberry, respectively, a chromosome map (Fig. 1) was drawn based on genome annotation (Wu et al., 2012). In pear, two EIN3/EIL genes were located on chromosome 2, and one gene on chromosome 3, 8, 11 and 15, respectively, with the remaining genes localized on different scaffold regions (Fig. 1C). In peach, two EIN3/EIL genes were found on chromosome 2 and 6, respectively (Fig. 1D). In mei, two EIN3/EIL genes were distributed on chromosome 1 and 5, respectively, with the remaining one localized on a scaffold region (Fig. 1B). In strawberry, two EIN3/EIL genes were distributed on chromosome 1 and 7, respectively, with the remaining one localized on chromosome 3 (Table 1 and Fig. 1A).

Phylogenetic analysis of EIN3/EIL genes

The phylogenetic tree containing EIN3/EIL gene homologs from a variety of species, including Arabidopsis thaliana, rice, banana, sorghum, maize, Brachypodium distachyon, Thellungiella parvula, and four Rosaceae species, was also constructed (Fig S1). As shown in the phylogenetic tree, most EIN3/EIL genes from four Rosaceae species were clustered together. To further understand the evolutionary history of EIN3/EIL genes in Rosaceae, phylogenetic analysis was carried out using the neighbor joining (NJ) method. As shown in Fig. 2, 24 EIN3/EIL sequences were divided into two subfamilies, designated as A and B, which contained four classes (Classes A1, A2, B1 and B2). Classes A1, B1 and B2 were composed of the EIN3/EIL genes from the four species (pear, peach, mei and strawberry), while the Class A2 contained the members only from pear. According to our study, a whole genome duplication event happened 30–45 million years ago in pear, but not in peach, mei and strawberry (Wu et al., 2012). This result suggested the probable reason for occurrence of Class A2 genes in pear. Remarkably, EIN3/EIL genes from peach and mei showed higher similarity with each other according to genetic distance, which was consistent with a previous study reporting that the closer relationship between peach and mei versus peach and pear/strawberry (Cao et al., 2016a; Cao et al., 2016b; Cao et al., 2016d).

Figure 2 Phylogenetic tree of EIN3/EIL proteins from pear, peach, mei and strawberry.

The neighbor-joining (NJ) tree was constructed by using MEGA 5 software. The neighbor-joining (NJ) tree was constructed by using MEGA 5 software (1,000 bootstrap replicates). The different colors suggest the different species background for each EIN3/EIL protein. Gene names are listed in Table 1. The scale bar represents 0.1 amino acid changes per site.

Each of the four Rosaceae species contributed at least one member of the EIN3/EIL gene to each class, with the exception for Class A2 (Fig. 2). Therefore, we deduced that EIN3/EIL genes had rapidly been duplicated before these dicotyledon species diverged. However, only the EIN3/EIL genes in class A2 revealed an internal duplication. In addition, we identified three pairs of orthologous genes among the EIN3/EIL genes: PmEIL3 and PpEIL1, PpEIL2 and PmEIL4, PpEIL3 and PmEIL5 based on the phylogenetic analysis.

Structural analysis of EIN3/EIL genes

Previous studies have suggested that gene structural diversity is the primary resource for the evolution of multigene families (Cao et al., 2016c; Leitch & Leitch, 2013; Mercereau-Puijalon, Barale & Bischoff, 2002). To characterize the structural diversity of the EIN3/EIL gene family, exon-intron organization of each EIN3/EIL gene was analyzed. As shown in Fig. 3A, most genes did not contain introns, such as FvEIL1, PbEIL1, PmEIL3 and PpEIL4 et al. Furthermore, PbEIL7 contained eight introns, followed by FvEIL3 (five), whereas FvEIL5 had three introns, PmEIL2 had two introns and eight EIN3/EIL genes contained one intron (Fig. 3A). These results implied that the intron/exon loss and acquire has occurred in the evolution of the EIN3/EIL gene family, which may be able to explain the functional divergence of closely related EIN3/EIL genes. In the present study, the gene structures of the EIN3/EIL homologous gene pairs were investigated. We found that the exon number of two gene pairs (PbEIL2/PbEIL8 and PmEIL1/PmEIL2) had changed. Further analysis indicated that PbEIL8 and PmEIL2 obtained one exon during evolution, while PbEIL2 and PmEIL1 lost one exon. These diversities might be due to single intron loss or obtain events during evolution.

Figure 3 Gene structure (A) and conserved motif compositions (B) of EIN3/EIL genes in Rosaceae species.

Untranslated regions (UTRs), introns and exons are represented by blue boxes, thin lines and green rectangles, respectively. Note that the gene or protein lengths can be estimated by using the scale at the bottom.

Because 24 EIN3/EIL genes did not have high similarity, MEME web server was used to find conserved motifs. Subsequently, we identified 20 conserved motifs, which were shown in Table S2 and Fig. 3B. The Pfam and SMART databases were used to annotate the individual of the putative motifs. Motif 1 and motif 2 were identified to encode a conserved EIN3/EIL domain (Chang & Shockey, 1999; Chao et al., 1997), whereas the remaining motifs did not get function annotation. Most EIN3/EIL proteins have motifs 1, 2, 4, 7, 10 and 19. In addition, several proteins from clade B2 contained unique motif 12, which might imply its specific functions. Remarkably, most of the closely related EIN3/EIL proteins in the same clade exhibited similar motif compositions (e.g., PmEIL5/PpEIL3 and PbEIL3/PbEIL5) indicating their functional similarity among these EIN3/EIL proteins. Summarily, the similarity in motif distribution and exon-intron structure of most EIN3/EIL proteins supported the results from phylogenetic analysis of the EIN3/EIL genes, whereas the differences of the related characteristics in the different classes indicated that their functions were diversified.

Conserved microsynteny of EIN3/EIL genes in the four Rosaceae species

Based on the whole-genome data, species microsynteny can be used to identify the location of orthologous genes and/or paralogous genes (Cao et al., 2016d; Lin et al., 2014). To identify the homologous genes (orthology or paralogy) within the EIN3/EIL genes from four Rosaceae species (pear, peach, mei and strawberry), as well as their evolutionary history, microsynteny analysis was performed. By pairwise comparisons of flanking sequences in the chromosomal regions containing EIN3/EIL genes, three or more pairs were present in this region, which were considered as either conserved microsynteny or high levels of microsynteny.

In this study, a total of 55 flanking sequences containing EIN3/EIL genes could be assembled into 28 regions and divided into seven microsynteny groups. It was supposed that EIN3/EIL genes from the same group should evolve from the most recent common ancestor. Based on this criterion, orthology and/or paralogy relationships, as well as their evolutionary origins, were detected among EIN3/EIL genes of the four Rosaceae species. Nine, five, four and four out of the seven microsynteny groups are from pear, peach, mei and strawberry, respectively (Fig. 4). In class B2, two gene pairs (PbEIL1 and PbEIL9, PbEIL9 and PbEIL10) with both a higher level of microsynteny and a noticeable inverted duplication, were identified. Interestingly, it was also found that some duplication rules in several regions were in disorder, such as FvEIL1 and FvEIL2, FvEIL1 and PpEIL4 (Fig. 4). Similar microsynteny was identified in other classes with a concordant inverted microsynteny (Fig. 4). In addition, according to the constructed phylogenetic tree, conservation of microsynteny between different families appeared gradually. However, some flanking genes in each microsyntenic group were not conserved, indicating that they arose later than this duplication event (Fig. 4). Furthermore, we only identified four pairs of intraspecies microsynteny groups from pear (PbEIL1 and PbEIL9, PbEIL3 and PbEIL5, PbEIL4 and PbEIL6, PbEIL9 and PbEIL10), but peach, mei and strawberry were excluded (Fig. 5). This difference might result from the expansion of pear EIN3/EIL genes. However, no similar gene expansion was identified in peach, mei and strawberry. Some previous studies has hypothesized that transcription factors should be generally and preferentially retained after genome duplications (Blanc & Wolfe, 2004), with a lower frequency of tandem duplication events in a number of transcription factors (Freeling, 2009). Additionally, genes from whole-genome duplication events are more easily retained into genomes. With the stoichiometric relationships, these genes were strongly retained by stabilizing selection (Lynch & Conery, 2000). Our results were not only consistent with this hypothesis, but was also strong evidence for it.

Figure 4 Interspecies microsynteny related to EIN3/EIL families in four Rosaceae.

The relative positions of all flanking protein-coding genes were defined by anchored EIN3/EIL genes, highlighted in red. The gene’s orientations are shown as triangle, with gray lines corresponding to chromosomal segments.

Figure 5 Intraspecific microsynteny related to EIN3/EIL families with the same species.

The relative positions of all flanking protein-coding genes were defined by anchored EIN3/EIL genes, highlighted in red. The gene’s orientations are shown as triangle, with gray lines corresponding to chromosomal segments.

Table 2 The relative syntenic quality of EIN3/EIL genes in four Rosaceae plants.

	Clade A1	Clade A2	Clade B1	Clade B2	Average	
Pb-Pp			22.50%	44.07%	33.29%	
Pb-Pm			21.51%	41.18%	31.35%	
Pb-Fv			24.07%	28.26%	26.17%	
Pp-Pm	26.67%		40.00%	29.73%	32.13%	
Pp-Fv	10.26%				10.26%	
Pm-Fv	10.00%		26.32%	3.77%	13.36%	
					24.43%	
Notes.

The relative syntenic quality was estimated as twice the number of matches divided by the sum of the total number of genes in both conserved gene regions, based on the previous methods (Cannon et al., 2003; Cannon et al., 2006).

Subsequently, the quality of the synteny was estimated in four Rosaceae plants based on previous research methods (Cannon et al., 2003; Cannon et al., 2006). As shown in Table 2, the relative synteny quality of the EIN3/EIL genes from these Rosaceae four species genomes was 24.43% for orthologous regions. The highest value of synteny quality found between pear and peach was 33.29%. And the lower value of synteny quality was obtained between strawberry and peach (10.26%) and mei (13.26%) The relative synteny quality in the pear/mei and pear/strawberry syntenic regions was 31.35% and 26.17%, which was substantially lower than the 32.13% found in the pear/peach synteny blocks. Our results were essentially consistent with their evolutionary relationship (Xiang et al., 2017; Zhong et al., 2015).

Strong purifying selection for EIN3/EIL genes in four Rosaceae species

In general, Ks values can be used to estimate evolutionary data of the whole genome duplication events or segmental duplication events. Previous studies showed that pear had experienced two whole genome duplication events, including an ancient whole genome duplication (Ks ∼ 1.5–1.8) estimated at ∼140 MYA (Fawcett, Maere & Van de Peer, 2009) and a recent whole genome duplication (Ks ∼ 0.15–0.3) estimated at 30–45 MYA (Wu et al., 2012), while peach, mei and strawberry only experienced an ancient whole genome duplication event. Therefore, Ks values were applied to analyze the whole genome duplication or segmental duplication events in EIN3/EILs of four Rosaceae species. As shown in Table S3, the mean Ks values of each duplication pairs in the syntenic region were lists. In pear, we found the mean Ks values of EIN3/EIL gene pairs were 0.0363, 0.1717 and 0.2836, respectively. It was obvious that these duplications might be resulting from the latest whole genome duplication (30–45 MYA; Ks ∼ 0.15–0.3), but an ancient whole genome duplication (∼140MYA; Ks ∼ 1.5–1.8) in pear.

In addition, the Ka/Ks values are widely used to represent the gene selection pressure and evolution rate [40]: Ka/Ks value with >1 indicates positive selection with accelerated evolution, Ka/Ks < 1 indicates negative/purifying selection with the functional constraint, and Ka/Ks = 1 suggests that the genes are drifting neutrally. In this study, all paralogs was found with Ka/Ks ratios <1 (Fig. 6), indicating their purifying selection. Furthermore, to better understand the delineate regions of diversifying and purifying selection in the EIN3/EIL gene family, a sliding window analysis of the Ka/Ks values between paralogs was performed (Fig. 4); the EIN3/EIL domains in the N-termini exhibited stronger purifying selection compared with the whole gene regions (C-termini). These results suggested that the EIN3/EIL genes had undergone strongly purifying selection, especially for EIN3/EIL domains in the N-termini (Fig. 4Q). Overall, strong evolutionary constraints were involved in EIN3/EIL gene evolution, which may contribute to their functional stability. On the other hand, some parts of protein-coding genes had undergone positive selection, implying the generation of innovative gene functions.

Figure 6 Sliding window plots of duplicated EIN3/EIL genes in Rosaceae species.

The gray blocks indicate the positions of the EIN3/EIL domains. The window size is 150 bp, and the step size is 9 bp. The x-axis denotes the synonymous distances within each gene.

Figure 7 Expression profiling of pear PbEIL genes in eight samples from root, stem, leaves and fruits in several development stages.

The expression profile data was obtained with qRT-PCR experiments. Blue and red colors indicate low-expression and high-expression, respectively.

Expression profiles analysis of PbEIL genes in different tissues

To increase our understanding of the potential functions of pear EIN3/EIL genes during development, qRT-PCR analysis was carried out to determine the expression profiles of ten PbEIL genes in different tissues. As shown in Fig. 7 and Table S4, ten pear EIN3/EIL genes showed significantly different tissue-specific expression patterns in eight samples from root, stem, leaves and fruits in several development stages. Among the ten pear EIN3/EIL genes, three (PbEIL5, PbEIL6 and PbEIL10) showed the highest transcript accumulation in the leaves, three (PbEIL2, PbEIL3 and PbEIL9) in 145 DAF (days after flowering), two (PbEIL1 and PbEIL4) in 79 DAF, and one (PbEIL7) in the roots. Additionally, the duplication gene pairs showed different expression patterns; for example, PbEIL4 was highly expressed in 79 DAF, while its duplication gene, PbEIL6, was expressed at a high level in the leaves. Thus, the pear EIN3/EIL duplicates resulting from recent whole genome duplication have different expression patterns in several different tissues, indicating subfunctionalization after duplication. At the same time, this phenomenon was also observed among other EIN3/EIL duplication genes (Jourda et al., 2014).

Conclusions

In this study, we identified 24 EIN3/EIL genes from four Rosaceae species (pear, peach, mei and strawberry). Subsequently, a systematic analysis, including their chromosomal location, evolutionary relationship, conserved microsynteny, gene structure and sliding window, was carried out. According to phylogenetic analysis, the EIN3/EIL genes divided into four classes. Remarkably, high level of microsynteny of the EIL/EIN3 family in Rosaceae was found, indicating that the genome duplication plays a key role in the expansion of the EIL/EIN3 genes in the Rosaceae. In these EIL/EIN3 genes, all paralogs have experienced purifying selection, especially the EIL/EIN3 domains in the Rosaceae. Furthermore, the expression profiles of the PbEIL genes suggested that the recent whole genome duplication derived genes show indications of subfunctionalization. These results may help promote the extrapolation of EIL/EIN3 gene functions in future.

Supplemental Information

Supplemental Information 1 Supplemental files

Click here for additional data file.

Supplemental Information 2 List of EIN3/EIL genes identified in pear, peach, yangmei and strawberry

Click here for additional data file.

We would like to thank Muhammad Abdullah for his careful reading and helpful comments on this manuscript. We extend our thanks to the reviewers and editors for their careful reading and helpful comments on this manuscript.

Additional Information and Declarations

Competing Interests

Author Contributions

Data Availability

The authors declare that they have no conflict of interest.

Yunpeng Cao conceived and designed the experiments, performed the experiments, analyzed the data, contributed reagents/materials/analysis tools, wrote the paper, prepared figures and/or tables, reviewed drafts of the paper.

Yahui Han conceived and designed the experiments, analyzed the data, wrote the paper.

Dandan Meng analyzed the data, wrote the paper, reviewed drafts of the paper.

Dahui Li conceived and designed the experiments, prepared figures and/or tables.

Qing Jin and Yi Lin reviewed drafts of the paper.

Yongping Cai conceived and designed the experiments, analyzed the data.

The following information was supplied regarding data availability:

The raw data has been supplied as Supplementary File.

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
