# Peer review of "Genome-wide analysis suggests high level of microsynteny and purifying selection affect the evolution of EIN3/EIL family in Rosaceae"

_PeerJ, doi:10.7717/peerj.3400_

## Round 0.1 · original submission · Major Revisions

Please provide an improved manuscript according to the reviewers' suggestions. Please, explain all improvements and changes in the manuscript in a rebuttal letter.

Reviewer 1 ·

Basic reporting

Unfortunately the authors find it very difficult to effectively communicate the information generated due to the major flaws in sentence construction and grammatical errors which can be found throughout the manuscript (right from abstract to conclusion).

A few grammatical mistakes are listed below
Abstract 2nd sentence “Although the EIN3/EIL gene family has been studied in several plant species, no comprehensive study of the EIN3/EIL gene family in Rosaceae.” Is incomplete
Change the sentence “In addition, whole genome duplication events and tandem duplication events played a key role in gene expansion of EIL/EIN3 gene fam ily” to

Line 44-46, change the structure
46-47- “and” is repeated, chromosome spelling incorrect
Line 50-52- not clear , what the authors mean to say
Line 62- not clear
Line 69- sentence break
Line 73- 75- change structure
Line 81 Spelling for “Institude”
Change “The genome of four Rosaceae species was obtained from the genome sequence websites, respectively” to The genome of four Rosaceae species was obtained from their respective genome sequence websites.

Line 100- sentence break (sentence should not start with “and”
Line 103- “annotate the structural motif” is correct

Line 149 Spelling/grammer for “was draw”
Line 271: Spelling for “resultts”
Line 271: Correct the sentence “may help promote”
Since the manuscript contains numerous grammatical mistakes, i stopped pointing out the language corrections at this stage. The entire manuscript may be revised for language correction with the help of an expert in English language.

Experimental design

The methodology adopted and the software’s used were good enough to generate meaningful information.

Validity of the findings

The findings are interesting and will be useful if proper wet lab experiments follows.
conclusions section especially on purifying selection is vague

Additional comments

The authors have done a good job by making use of the available genomic resources of four Rosaceae species to systematically analysis the EIN3/EIL genes present in them which are essential transcription factors in the ethylene signalling of higher plants.

Altogether, the idea which has gone behind the study and the approach the authors have taken to arrive in to the conclusions (though little vague) is appreciable.

I feel that the manuscript may be more insightful with the inclusion of the following points.

1) The authors could have discussed a bit more on the functional role of the EIN3/EIL gene family in general and the variations if reported in other crops.
2) The reason for selecting EIN3/EIL gene family for phylogenetic and microsynteny analysis ( There and several such gene families which shares many portions of their genome among related species in Rosaceae family as well as other families.
3) The accuracy of the statement on purifying selection based on a simple comparative genome analysis study may be justified properly. Eexperimental evidences based on evolunationary studies are generally carried out to arrive in to such a conclusion.
4) If possible the tissue-specific expression profiles of the different genes in four Rosaceae species may be included. Transcriptome data from different tissues may be used if available (Not mandatory if generating wet lab data is beyond the scope of this study)

Technically the paper is sound and the data quality is good. Since the manuscript requires mainly language corrections, I am recommending minor revisions.
The manuscript may be accepted for publication after revising it as suggested.

Reviewer 2 ·

Basic reporting

• The manuscript entitled “Genome-wide analysis suggests high level of microsyntheny and purifying selection affect the evolution of EIN3/EIL family in Rosaceae” is an interesting study of the evolution of a gene family using a comparative analysis in 4 plant species at genome scale.

• A better description of the importance of the studied family could improve the general comprehension of this manuscript

• Please, be careful! Could you check and correct your manuscript?
For example:
- Lines 38,52, 70, 71, 75, 85, 102, 138: EIN3/EIL in italics
- Line 47: typo for “chromosomesn”
- Lines 60, 204, 222: be careful with spaces
- Line 100: sequence and a diagram […]
- Line 103: used to annotate each structural motif
- Line 123: Furthermore, to […]
- Line 134: typo for “SMATR database”
- Line 134: typo for “resultts”

• Fig.1: What is the unit of scale bar?
• Fig.2: Could you add a legend for the scale bar?
• Fig.3: Could you add scale bars for lengths?
In addition, it's very difficult to differentiate motifs labeled in green. Could you used others colors (e.g. grey and/or black)?
• Fig.6: Could you add the legend for axis of abscissas??
• Table S2: Could you explain the s.d. (standard deviation?) acronym in a legend under the table

Experimental design

• The experimental procedure is quite standard, and is appropriate for this study
• Methods are described with sufficient detail & information to replicate

Validity of the findings

• The experimental setting has to be extent to other plant genomes (e.g. grasses and solanaceae) to have a robust phylogeny reconstruction.

• I am very surprised: no comparison was performed with a previous similar study (Jourda C, Cardi C, Bocs S, Garsmeur O, D’Hont A, Yahiaoui N (2014) Expansion of banana (Musa acuminata) gene families involved in ethylene biosynthesis and signalling after lineage‐specific whole‐genome duplications, New Phytologist: 202 (3), 986-1000. For example, the number of EIN3/EIL subfamilies are differents.

• A discussion comparing models explaining gene retention after duplication could improve your manuscript using for example:
- De Smet R, Van de Peer Y. 2012. Redundancy and rewiring of genetic networks following genome-wide duplication events. Current Opinion in Plant Biology 15: 168–176.
- Force A, Lynch M, Pickett FB, Amores A, Yan Y, Postlethwait J. 1999. Preservation of duplicate genes by complementary, degenerative mutations. Genetics 151: 1531–1545.
- Hittinger CT, Carroll SB. 2007. Gene duplication and the adaptive evolution of a classic genetic switch. Nature 449: 677–681.
- He X, Zhang J. 2005. Rapid subfunctionalization accompanied by prolonged and substantial neofunctionalization in duplicate gene evolution. Genetics 169: 1157–1164.
- Bergthorsson U, Andersson DI, Roth JR. 2007. Ohno’s dilemma: evolution of new genes under continuous selection. Proceedings of the National Academy of Sciences 104: 17004–17009.
- Birchler JA, Riddle NC, Auger DL, Veitia RA. 2005. Dosage balance in gene regulation: biological implications. Trends in Genetics 21: 219–226.
- Birchler JA, Veitia RA. 2007. The gene balance hypothesis: from classical genetics to modern genomics. The Plant Cell 19: 395–402.
- Bekaert M, Edger PP, Pires JC, Conant GC. 2011. Two-phase resolution of polyploidy in the Arabidopsis metabolic network gives rise to relative and absolute dosage constraints. The Plant Cell 23: 1719–1728.

In addition, experimental data (e.g. qPCR) could help to support your evolutionary hypothesis (lines 224-229).

Additional comments

• Titles for sections of results could be more exciting

Reviewer 3 ·

Basic reporting

no comment

Experimental design

no comment

Validity of the findings

no comment

Additional comments

This manuscript presents a study about the gene family of EIN3/EIL in Rosaceae. Authors have done lots of analyses, the study is interesting in general.
There are some issues need to be addressed:
1) Usage of strange terms to show the information. For example, in table 2, what does the average relative syntenic quality indicate? Is it the similarity among homologs?
2) More detailed description about the figure is needed in the figure legends section.
3) On Figure 3, it will be better to show the motif names nearby their symbols, instead of only mark a number nearby the symbol, too less information.
4) On Figure 5, this figure actually only compared EIN3/EIL genes with the same species, instead of “in four Rosaceae”.
5) More importantly, I would strongly suggests authors to pay much more attention to the most significant finding of this study, however, in the present format, which is not clear to me.

---

## Round 0.2 · accepted · Accept

Your manuscript is ready for acceptance. I attach an annotated version with some slight language corrections in the Abstract and Introduction.

Reviewer 1 ·

Basic reporting

OK

Experimental design

OK

Validity of the findings

OK

Additional comments

The authors have satisfactorily responded to all my questions and made the necessary changes to the manuscript. It may be considered for publication in PeerJ